# Bacteriological analysis and antibiotic resistance in patients with diabetic foot ulcers in Dhaka

**Poulomi Baral[1]°, Nafisa Afnan[1]°, Maftuha Ahmad Zahra[1]°, Baby Akter[2]‡, Shek Rabia Prapti[2]‡, Mohammed Muazzam Hossan[3], Fahim Kabir Monjurul Haque[1]***

1 Microbiology Program, Department of Mathematics and Natural Sciences, BRAC University, Dhaka, Bangladesh, 2 Biotechnology Program, Department of Mathematics and Natural Sciences, BRAC University, Dhaka, Bangladesh, 3 Shaheed Suhrawardy Medical College, Sher-E-Bangla Nagar, Dhaka, Bangladesh

☯ These authors contributed equally to this work.
‡ BA, and SRP also contributed equally to this work.
* fahim.haque@bracu.ac.bd

## Abstract

The primary objective of this study was to isolate bacteria from diabetic foot ulcers and subsequently assess their antibiotic resistance capabilities. Seventy-five patients diagnosed with diabetic foot ulcers were investigated. A number of these patients (97.33%) had type 2 diabetes, with a significant proportion of them having been diagnosed for 1–5 years (29.33%). Notably, a substantial number of these individuals were on insulin usage (78.66%). Among the patients under examination, 49.33% reported having no use of tobacco products, alcohol, or betel leaf. The ulcers analyzed in this study were classified into grades 1–5 according to the Wagner scale. Wagner grade 2 diabetic foot ulcers had the highest number of culture-positive patients, at 33.33%. Pus samples collected from patients were cultured on selective media, and bacterial identity was confirmed by biochemical tests and polymerase chain reaction. A total of 141 isolates were isolated. Among the isolates, 82.97% gram-negative bacteria and 17.02% gram-positive bacteria were detected. *Klebsiella pneumoniae* was the most common isolate. *Proteus spp.*, *Escherichia coli*, *Pseudomonas aeruginosa*, and *Staphylococcus aureus* were also detected. Approximately 61.33% of the ulcers exhibited were polybacterial. In this study, it was observed that all bacterial isolates, except for *Proteus* spp., were primarily detected in patients classified under Wagner's grade 2. Moreover, antibiotic susceptibility was also tested on these 141 isolates. Among them, *Escherichia coli* showed the highest multidrug resistance, 81.81%. Most of the gram-negative bacteria were resistant to ampicillin. All of the gram-negative isolates exhibited high levels of susceptibility to piperacillin-tazobactam, and these levels were *Klebsiella pneumoniae* (97.56%), *Pseudomonas aeruginosa* (95.24%), *Escherichia coli* (81.82%), and *Proteus spp.* (80%). On the other hand, gram-positive *Staphylococcus aureus* mostly showed sensitivity towards vancomycin and norfloxacin (79.17%).

**Data Availability Statement:** All relevant data are within the paper and its Supporting Information files.

**Funding:** The author(s) received no specific funding for this work.

**Competing interests:** The authors have declared that no competing interests exist.

## Introduction

Foot ulcers from diabetes are one of the greatest challenges for diabetic patients. People who have diabetes with a prior record of foot ulcers face a mortality risk that is two times higher than that of those without a foot ulcer history [1]. Patients who have diabetes are estimated to have a 25% lifetime probability of developing diabetic foot ulcers (DFU) [2]. A full-thickness wound below the ankle in a person with diabetes is known as DFU. According to the World Health Organization, DFU is an open sore wound of the foot related to neuropathy, ischemia, and infection. Worldwide, it is anticipated that 9.1 million to 26.1 million diabetic individuals may develop foot ulcers based on the International Diabetes Federation's prevalence [1].

Diabetes is the primary reason for DFU, and the global prevalence of DFU ranges from 3% to 13%. In North America, the prevalence of DFU is 13%, and in Africa (7.2%), Asia (5.5%), Europe (5.1%), and Oceania (3%) [3]. Furthermore, in various parts of Ethiopia, the prevalence of diabetic foot ulcers was found to be 14.8% and 13.6% in the regions of Arba Minch and Gondar [4–7]. Patients who suffer from DFU have a higher morbidity and mortality rate worldwide, leading them to spend a significant amount of time in the hospital and develop a risk of limb amputation [8]. According to the International Diabetes Federation, DFU leads to the removal of at least one limb every 30 seconds worldwide [9]. A study conducted in Australia and the US showed that a limb is amputated every three hours, and 80,000 amputations annually due to DFU [10, 11]. During diabetes,14–24% of people undergo a lower extremity amputation because of DFU [12]. According to a recent study conducted on adults ranging in age from 20 to 79 years old revealed that there was a total of 537 million patients with diabetes in 2021, and it is expected that this number will climb to 643 million by 2030 and to 784 million by 2045 according to [13]. Infections can be caused by various bacteria, including gramnegative and gram-positive bacteria [14]. A study stated that anaerobic bacteria may also be present in more severe and deeper phases of the DFU [15]. In several previous studies, the most prevalent bacterium detected was *Staphylococcus aureus* [14–22].

Antibiotic resistance is an emerging global concern, and studies have reported the presence of antibiotic-resistant bacteria in DFU. The World Health Organization (WHO) has recognized it as a serious public health threat [23]. Therefore, the detection of bacteria and their susceptibility pattern is necessary to ensure proper antibiotic selection [14].

In Bangladesh, DFU is becoming highly prevalent. According to a trend study, in Bangladesh, the incidence of diabetes increased from 4% in 1995–2000 to 5% in 2001–2005 and gradually rose to 9% in 2006–2010 [24]. Furthermore, another study showed that 9.7% of the Bangladeshi adult population aged 35 or older has a prevalence of diabetes, and 22.4% are prediabetic [25].

Few studies on bacteriology and antibiotic resistance to DFU have been done in Bangladesh [26, 27]. Nevertheless, these investigations were carried out a considerable time in the past. So, it would be valuable to reexamine them to determine whether the same bacterial isolates continue to be the primary culprits behind DFU. Additionally, assessing whether antibiotic resistance patterns remain consistent or have undergone any alterations would be of great significance. Furthermore, this study classifies bacterial distribution according to Wagner's scale, a method employed by only one previous study conducted in Bangladesh [26]. Besides, there is a lack of knowledge related to the DFU's associated risk factors among Bangladeshi people. In this study, our main objective was to clinically identify the bacterial sources of individuals with diabetic foot ulcers in Bangladesh while also evaluating their antibiotic resistance. Additionally, we explored the risk factors associated with DFU among the Bangladeshi population. The insights gained from this study will contribute to better management of future cases and help in developing a guideline for patient treatment. We believe that through our

comprehensive examination, we have successfully achieved this aim, providing valuable insights for both clinical practice and public health initiatives.

## Methods

### Study design

This cross-sectional study collected patient samples from two tertiary care hospitals in Dhaka, Bangladesh: Ibrahim Diabetic Foot Care Hospital and Shaheed Suhrawardy Medical College and Hospital. The patient samples were collected from 25 May 2022 to 30 September 2022. In total, 75 DFU patients' pus specimens were collected. Participants of both sexes, aged 30–80 years old, were interviewed regarding their medical histories of diabetic foot ulcers.

### Population and sample

This study was conducted after gaining informed consent from study patients. To meet the study's eligibility criteria, the participants were required to exhibit indications of a diabetic-related bacterial foot ulcer. A diabetic foot infection is essentially defined as an infection below the ankle in an individual with diabetes mellitus. This category encompasses various conditions such as paronychia, cellulitis, myositis, abscesses, necrotizing fasciitis, septic arthritis, tendonitis, and osteomyelitis. Nevertheless, the most prevalent and typical manifestation is the infected diabetic "mal perforans" foot ulcer. The clinical diagnosis of infection was established when two or more of the following criteria were met: local swelling or firmness, erythema extending beyond 0.5 cm from the wound, localized tenderness or discomfort, elevated local temperature, and the presence of purulent discharge [28]. Individuals with ulcers were classified into grades 1–5 based on the Wagner scale, which assesses a range of factors, including wound depth, penetration, the determination of osteomyelitis or gangrene presence, as well as the extent of tissue necrosis. This scale is specifically designed to evaluate the depth of ulcer wounds [29]. The scale is mentioned in detail in S1 Table.

### Sample collection

Each specimen was collected from a cleaned wound, where sterile physiological saline and gauze were used for cleaning, and debridement was performed to eliminate necrotic tissue, foreign substances, calluses, and undermined wound edges. It is important to note that no antimicrobial agents, such as alcohol or iodine, were applied to the wound before specimen collection [30]. The collection process followed the Levine technique, which involves the swab's rotational movement across a wound area of 1 cm$^2$ for a duration of 5 seconds. Sufficient pressure was applied in this technique to draw fluid from the inner part of the wound [31]. Subsequently, the collected specimens were carefully placed into tubes containing Amies transport medium. Following that, all samples were meticulously transported, maintaining a cold chain from the hospitals to the microbiology laboratory for aerobic culturing within two hours from the time they were collected. The whole collection process was done under the supervision of medical professionals.

### Data collection

The data was collected using a printed questionnaire. While preparing the questionnaire, we considered the demographic characteristics, behavioral factors, and clinical characteristics of the DFU patients. Each patient was interviewed in order to gather comprehensive information regarding their foot ulcer condition.

## Statistical analysis

This study used SPSS 18.0 software for Windows (IBM) for statistical analysis.

## Ethical considerations

The study was conducted in accordance with the Declaration of Helsinki, and ethical approval was obtained from the Institutional Review Board of BRAC University. Informed written consent was collected from each study participant prior to data collection. For the convenience of the participants, the consent form was prepared in both Bengali and English. For patients who were considered mentally unstable, ethical permission was obtained from their legal guardians. No minor was included in this study. During data collection and analysis, the respondents' privacy and the confidentiality of the information were strictly ensured. The interview was entirely voluntary, and the information gathered was used anonymously.

## Bacterial culture

All the samples were diluted using physiological saline, and the specimens were cultured on selective and differential media employing the spread plate technique. Subsequently, the plates were incubated at 37˚C for a period of 24–48 hours, and colony morphology was observed. S2 Table provides information on the selective media used and the corresponding colony morphology.

## Biochemical tests

Further bacterial identification was made using biochemical tests. These tests include the catalase test, oxidase test, triple sugar iron agar test, motility indole urease test, methyl red test, Voges Proskauer test, and citrate utilization test. S3 Table outlines the criteria for interpreting biochemical tests. For the suspected *Staphylococcus aureus* isolates, confirmation was achieved through hemolysis and coagulase testing. For suspected *Proteus spp*., isolates were subjected to additional subculturing on Blood Agar, which allowed the observation of swarming colonies. Similarly, suspected *Escherichia coli* isolates were also subjected to additional subculturing on EMB Agar. Subsequently, all bacterial isolates except *Proteus spp*. underwent Polymerase Chain Reaction (PCR) for precise identification.

## DNA extraction and amplification

The isolates' genomic DNA was extracted by following the boiling method. The isolates were cultured in Luria Bertani broth, and the broth was incubated at 37˚C overnight. After transferring 700μl of the bacterial culture to microcentrifuge tubes, it was centrifuged at 13000 rpm for 10 minutes. After that, the pellet was washed with 300μl phosphate buffer solution (PBS) and then centrifuged at 13000 rpm for 5 minutes. The pellet was then suspended in 200μl of TE buffer and boiled for 15 minutes at 100˚C in a water bath, followed by 10 minutes of cold shock. The cell debris was precipitated after the cell suspension was centrifuged at 13000 rpm for 5 minutes. Lastly, the final supernatant containing the DNA was collected and stored at -20˚C.

To prepare the PCR solution, 4μl of template DNA, 12.5 μl of Master Mix, 4.5μl of nuclease-free water, 2μl of forward primer, and 2μl of reverse primer were combined to achieve a final solution volume of 25 μl for each bacterial sample. The PCR products underwent electrophoresis on a 2% agarose gel with 1X TAE buffer that was stained with ethidium bromide. Subsequently, the products were visualized under a UV transilluminator. Detailed information on primer sequences, PCR thermocycler conditions, and amplicon sizes can be found in S4 Table.

## Antimicrobial susceptibility test

To determine the drug resistance pattern of these isolated bacteria, the Kirby-Bauer disc diffusion method was used to conduct an antibiotic susceptibility test. The zone diameters of the selected antimicrobial agents were interpreted following the Clinical Laboratory Standards Institute (CLSI) 2018 guidelines. In S5 Table, a compilation of antibiotics used, along with their corresponding groups, effectiveness, disc potency, and interpretive criteria is given. Single-drug resistance was characterized as resistance to a sole antibiotic class. Multidrug resistance (MDR) was defined as resistance to at least one agent in three or more antimicrobial categories, following the criteria established by the Clinical Laboratory Standards Institute (CLSI), European Committee on Antimicrobial Susceptibility Testing (EUCAST), and the United States Food and Drug Administration (FDA) [32].

## Flowchart of the methods used in this study

The methods that were followed in this study are shown in a flowchart in Fig 1.

# Results

## Demographic characteristics

This study included a total of 75 participants, with 44 individuals sourced from Shaheed Suhrawardy Medical College and Hospital, Dhaka and the remaining 31 from Ibrahim Diabetic Foot Care Hospital, Dhaka. Among these participants, 41 (54.66%) were male, while 34 (45.33%) were female. The majority of the study's subjects fell within the age range of 30 to 80, with an average age of SD±7.5 years. In terms of educational background, 23 (30.66%) of the participants had no formal education. Furthermore, a significant proportion of the participants, specifically 53 (70.66%), hailed from urban areas. A summary of the demographic characteristics of the study's patients is presented in Table 1.

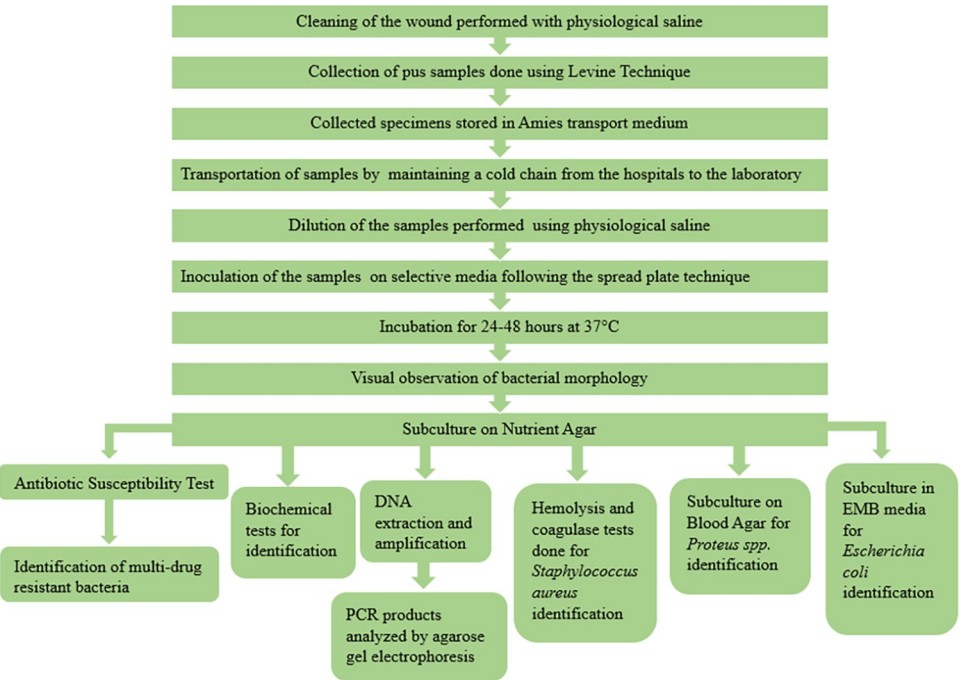

**Fig 1. Flowchart detailing the methods used in this study.**

**Table 1. Demographic characteristics of patients.**

| Characteristics | Categories | Frequency | Percentage |
|---|---|---|---|
| **Number of participants at study sites** | Shaheed Suhrawardy Medical College and Hospital | 44 | 58.66 |
| | Ibrahim Diabetic Footcare Hospital | 31 | 41.33 |
| **Gender** | Men | 41 | 54.66 |
| | Women | 34 | 45.33 |
| **Age (in years)** | 30–35 | 2 | 2.66 |
| | 36–40 | 4 | 5.33 |
| | 41–45 | 10 | 13.33 |
| | 46–50 | 25 | 33.33 |
| | 51–55 | 7 | 9.33 |
| | 60 | 13 | 17.33 |
| | 61–65 | 6 | 8 |
| | 66–70 | 4 | 5.33 |
| | 71–75 | 2 | 2.66 |
| | 76–80 | 2 | 2.66 |
| **Employment Status** | Employed | 38 | 50.67 |
| | Unemployed | 37 | 49.33 |
| **Education** | No formal educational background | 23 | 30.66 |
| | Class 1-Class 10 | 24 | 32 |
| | [a] SSC | 12 | 16 |
| | [b] HSC | 7 | 9.33 |
| | Tertiary Education | 9 | 12 |
| **Locality** | Urban | 53 | 70.66 |
| | Rural | 22 | 29.33 |

[a] The Secondary School Certificate (SSC) examination in Bangladesh is a public examination conducted by the Boards of Intermediate and Secondary Education under the Ministry of Education. This examination marks the culmination of secondary education for students, typically taken after completing ten years of schooling.
[b] Students who pass the Secondary School Certificate (SSC) successfully proceed to enroll in a college for a two-year higher secondary education program. They must participate in the Higher Secondary Certificate public examination administered by the education boards.

## Behavioral risk factors and clinical factors

Among the patient population, a diverse range of behavioral habits was observed. Notably, 37 individuals (49.33%) reported abstaining from any behavioral risk habits, which encompassed the use of tobacco products (smokeless tobacco), smoking and alcohol. An equivalent percentage of 11 individuals (14.66%) both smoked and consumed betel leaf, while a small group of 3 patients (4%) were regular consumers of alcohol, with 4 individuals (5.33%) exhibiting a combination of tobacco, alcohol, and smoking habits. Additionally, a majority of 46 patients (61.33%) led sedentary lifestyles, while 29 patients (38.66%) actively engaged in physical activities, which included walking, physical exercise, and domestic labor. Furthermore, 53 patients (70.66%) maintained commendable foot hygiene practices, such as daily foot inspection, washing, and thorough drying, in contrast to 22 patients (29.33%) who refrained from these practices due to a lack of knowledge and awareness about DFU. The study participants overwhelmingly presented a diagnosis of type 2 diabetes, accounting for 73 (97.33%) of the cohort. On average, these participants had been living with diabetes for approximately 12.5 years. A significant majority, comprising 59 participants (78.66%), were under insulin treatment. Notably, wound localization data revealed that ulcers on the left leg were the most common, with 37 cases (49.33%), followed by ulcers on the right leg at 33 cases (44%). The overall result is presented in Table 2.

**Table 2. Study population characteristics.**

| Characteristics | Categories | Frequency | Percentage |
|---|---|---|---|
| **Tobacco consumption** | Betel leaf (paan) | 11 | 14.66 |
| | Gul, Jorda | 4 | 5.33 |
| | Tobacco (smoking) | 11 | 14.66 |
| | Alcohol | 3 | 4 |
| | Multiple [a] | 4 | 5.33 |
| | Neither Tobacco or Alcohol Consumption | 37 | 49.33 |
| **Physical Activity [b]** | Yes | 29 | 38.66 |
| | No | 46 | 61.33 |
| **Foot Self Care Practices [c]** | Maintained | 53 | 70.66 |
| | Not Maintained | 22 | 29.33 |
| **Type of diabetes** | Type-1 | 2 | 2.66 |
| | Type-2 | 73 | 97.33 |
| **Duration of Diabetes (Years)** | 1–5 | 22 | 29.33 |
| | 6–10 | 18 | 24 |
| | 11–15 | 15 | 20 |
| | 16–20 | 12 | 16 |
| | 21–25 | 6 | 8 |
| | 25–30 | 2 | 2.66 |
| **Insulin Use** | Yes | 59 | 78.66 |
| | No | 16 | 21.33 |
| **Mental Stability** | Normal | 66 | 88 |
| | Abnormal | 9 | 12 |
| **Wound Localization** | Both | 5 | 6.6 |
| | Right | 33 | 44 |
| | Left | 37 | 49.33 |
| **Hospital Stay** | 1–6 days | 19 | 25.33 |
| | 1 week | 14 | 18.66 |
| | 2–3 weeks | 16 | 21.33 |
| | 1 month | 13 | 17.33 |
| | 2–3 months | 13 | 17.33 |

[a] Consumed various tobacco products as well as alcohol

[b] Actively engaged in physical activities, which included walking, physical exercise, and domestic labor

[c] Maintained foot hygiene practices, such as daily foot inspection, washing, and thorough drying

## Additional complications

In this study, it was observed that a significant portion of the patients (40%) did not experience any additional complications due to DFU. A minor proportion of patients, approximately 14%, reported high blood pressure as an associated complication. Kidney complications were experienced by a smaller percentage of patients, around 9%. Gangrene was observed in a relatively small number of patients as well, about 7%. A minor portion of the participants, around 6%, reported experiencing eating disorders and low blood pressure. Liver complications, vomiting, and high blood sugar were also experienced by patients, with each of these complications occurring in approximately 3% of the cases. Heart complications were observed in around 2% of the patients. Complications such as high cholesterol, lung issues, gastritis, thumb pain, diarrhea, and severe pain were reported by a smaller number of patients, around 1%. This information is further detailed in Fig 2.

## Wagner scale-based classification of the DFU patients

The categorization of DFU patients using the Wagner classification involved assessing the depth or penetration of the wound, determining the presence or absence of osteomyelitis or gangrene, and evaluating the extent of tissue necrosis [29]. The full classification system is mentioned in S1 Table. In this study, diabetic foot ulcers in Wagner grade 2 displayed the highest prevalence of culture-positive patients, accounting for 33.33% of cases, followed by Wagner grade 3 with 19.85%, as depicted in Fig 3. As the Wagner score increased, there was a notable escalation in the incidence of infections caused by gram-negative bacteria. Moreover, an examination of bacterial classification based on the Wagner scale revealed a rising trend in ulcers characterized by polymicrobial development, indicating an association between ulcer severity and the likelihood of polymicrobial infection, as presented in Table 3.

## Isolated bacteria from diabetic foot ulcer

A total of 141 bacterial isolates were obtained from 75 DFU patients. The majority of these isolated bacteria were identified as gram-negative, accounting for 117 isolates (82.97%). The gram-negative bacterial species included *Klebsiella pneumoniae*, *Proteus spp.*, *Pseudomonas aeruginosa*, and *Escherichia coli*. In contrast, the only gram-positive bacteria identified was *Staphylococcus aureus*, with 24 isolates (17.02%) in total. Among the isolated bacteria, the most predominant bacteria was *Klebsiella pneumoniae* at 29%, followed by *Proteus spp.* at 21.27%, *Pseudomonas aeruginosa* at 17.02%, *Staphylococcus aureus* at 17.02% and *Escherichia coli* at 15.60%. The frequency of bacteria identified from DFU patients is shown in Table 4.

In the sample analysis, it was found that 18.67% of the samples exhibited monobacterial infection, 28% displayed polybacterial infection (caused by two organisms), 33.33% demonstrated polybacterial infection (caused by three organisms), while 20% showed no growth. The number of isolates detected in each sample is described in Table 5.

## Antibiotic susceptibility pattern of the isolates

A total of 141 randomly selected isolates underwent antibiotic susceptibility testing. The type and number of bacterial isolates that underwent the Kirby-Bauer Disc Diffusion test are

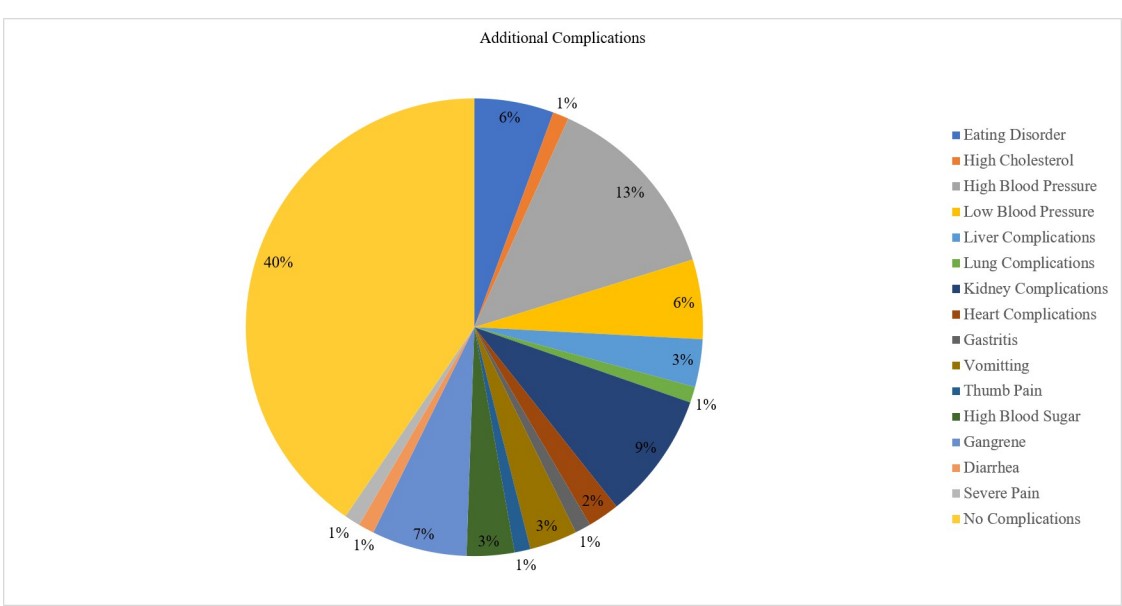

**Fig 2. Additional complications of diabetic foot ulcer patients.**

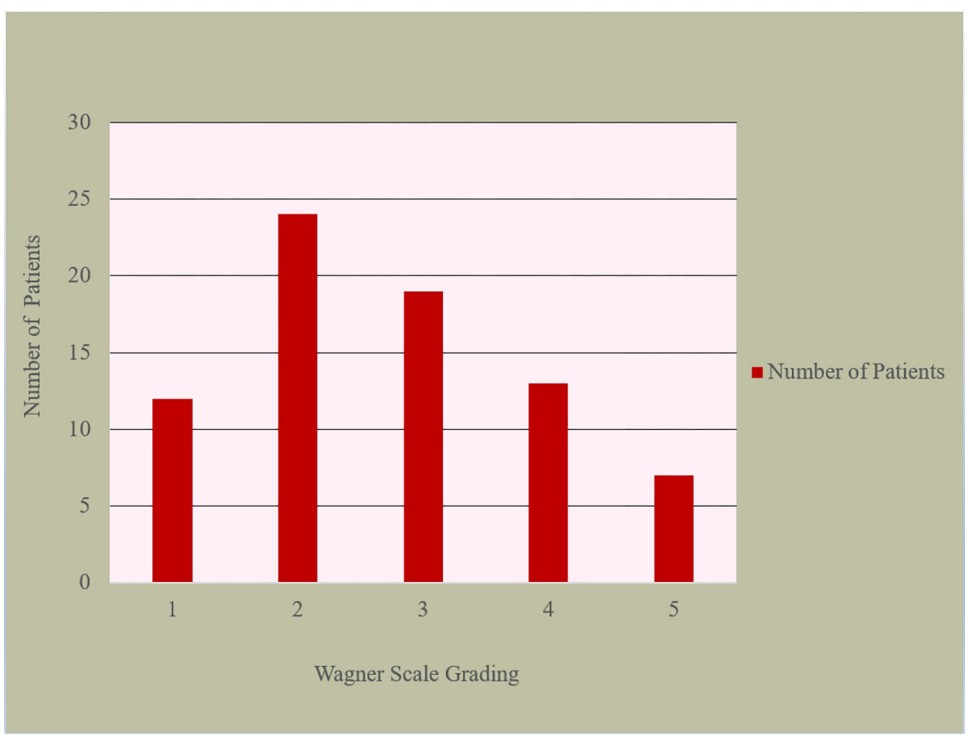

**Fig 3. Wagner scale grading of the patients.**

*Escherichia coli* (22 isolates), *Klebsiella pneumoniae* (41 isolates), *Pseudomonas aeruginosa* (24 isolates), *Proteus spp.* (30 isolates), and *Staphylococcus aureus* (24 isolates). The resistant isolates are shown in Table 6.

The majority of the *Escherichia coli* bacteria that were tested showed resistance to the antibiotics azithromycin (86.36%), ampicillin (81.82%), and colistin (77.2%). Medium sensitivity was seen for streptomycin (77.2%), meropenem (72.7%), amikacin (68.1%), tigecycline (59%), and imipenem (50%). Antibiotics like ampicillin (97.5%) and colistin (85.3%) showed a high level of resistance against *Klebsiella pneumoniae*, and a moderate level of resistance was noticed in cefepime (68.2%), tigecycline (60.5%) and azithromycin (58.5%). Amikacin (78%) showed the highest sensitivity. Streptomycin (68.2%) and imipenem (63.4%) showed moderate level of sensitivity. It was also found *Proteus spp.* was highly resistant to colistin (73.3%) and moderately resistant to tigecycline (63.3%) and azithromycin (60%). Piperacillin-tazobactam, ampicillin, streptomycin, and meropenem showed the highest sensitivity. Imipenem, cefepime, and amikacin showed moderate sensitivity for *Proteus spp.* For P*seudomonas aeruginosa*, it was noticed the highest level of resistance in ampicillin (95.2%), cefepime (70.8%), azithromycin

**Table 3. Wagner scale-based bacterial distribution.**

| Isolated Bacteria | Wagner Scale Grading | | | | |
|---|---|---|---|---|---|
| | Grade 1 | Grade 2 | Grade 3 | Grade 4 | Grade 5 |
| *Staphylococcus aureus* | 5 | 10 | 6 | 2 | 1 |
| *Klebsiella pneumoniae* | 8 | 14 | 6 | 7 | 6 |
| *Proteus spp.* | 6 | 7 | 7 | 6 | 4 |
| *Escherichia coli* | 3 | 8 | 3 | 4 | 3 |
| *Pseudomonas aeruginosa* | 2 | 8 | 6 | 4 | 4 |

**Table 4. Frequency of identified bacteria from diabetic foot ulcers.**

| Name of Bacterial Isolates | Frequency | Percentage |
|---|---|---|
| *Klebsiella pneumoniae* | 41 | 29 |
| *Proteus spp.* | 30 | 21.27 |
| *Pseudomonas aeruginosa* | 24 | 17.02 |
| *Escherichia coli* | 22 | 15.60 |
| *Staphylococcus aureus* | 24 | 17.02 |

(66.6%), and tigecycline (62.5%). *Staphylococcus aureus* was shown to have a high sensitivity to vancomycin (79.17%) and moderate sensitivity to linezolid (66.67%) in this study. However, it was also found that the isolated *Staphylococcus aureus* was highly resistant to cefepime (83.3%), ampicillin (83.3%), and azithromycin (79.1%), as well as moderately resistant to piperacillin-tazobactam (66.67%) and amikacin (62.5%).

A majority of the bacterial isolates exhibited resistance to multiple classes of antibiotics, indicating a high prevalence of multidrug resistance. *Escherichia coli* showed the highest level of multidrug resistance at 81.81%. This was followed by *Proteus spp.* (56.67%), *Staphylococcus aureus* (62.5%), *Pseudomonas aeruginosa* (58.33%), and *Klebsiella pneumoniae* (36%). The number of isolates showing multidrug resistance and single-drug resistance is described in Table 7.

## Discussion

As per the International Diabetes Federation, the diabetic population in Bangladesh consisted of 8.4 million adults in 2019, and this figure is projected to nearly double, reaching 15.0 million by 2045. In 2021, Bangladesh was 8th in terms of the highest number of adults (20–79 years) with diabetes, amounting to 13.1 million cases. According to IDF projections, Bangladesh is anticipated to ascend to the 7th rank by 2045 [9]. A study conducted in Bangladesh in 2019 showed that 45% of the test subjects with type 2 diabetes were likely to be affected by DFU.A 2019 study in Bangladesh revealed that 45% of individuals with type 2 diabetes, particularly the rural population, were susceptible to diabetic foot ulcers (DFU). As previously mentioned, microorganisms are the main cause of DFU, and for antimicrobial therapy, early diagnosis is essential [21]. In Bangladesh, as previously indicated, limited research has been conducted on bacteriology and antibiotic resistance in DFU in Bangladesh [26, 27]. These studies are now somewhat dated. Revisiting and re-evaluating these studies would provide valuable insights into whether the predominant bacterial isolates responsible for DFU have persisted over time. Equally important is the assessment of whether antibiotic resistance patterns have remained stable or have undergone changes.

In the current study, seventy-five people who were diagnosed with DFU participated. A significant number of participants experienced the onset of ulcers as minor blisters. These individuals assumed that the blisters would naturally heal within a short span, given their painless

**Table 5. Frequency of monomicrobial and polymicrobial infections in diabetic foot ulcers.**

| Nature of Microbial Infection | Frequency | Percentage |
|---|---|---|
| Monobacterial infection | 14 | 18.67 |
| Polybacterial Infection (caused by two organisms) | 21 | 28 |
| Polybacterial Infection (caused by three organisms) | 25 | 33.33 |
| No Growth | 15 | 20 |

**Table 6. Resistance profile of the isolated bacteria.**

| Antibiotics | Gram-negative bacteria | | | | Gram-positive bacteria |
|---|---|---|---|---|---|
| | *Proteus spp.* (N = 30) | *Klebsiella pneumoniae* (N = 41) | *Pseudomonas aeruginosa* (N = 24) | *Escherichia coli* (N = 22) | *Staphylococcus aureus* (N = 24) |
| **Macrolide** | | | | | |
| Azithromycin | 20 (66.67%) | 24 (58.53%) | 16 (66.67%) | 19 (86.37%) | 19(79.17%) |
| **Aminoglycoside** | | | | | |
| Amikacin | 11 (36.67%) | 9 (21.96%) | 12 (50%) | 7 (31.81%) | 15 (62.5%) |
| Streptomycin | 8 (26.67%) | 13 (31.7%) | 11 (45.83%) | 5 (22.72%) | 14 (58.33%) |
| **Glycylcyline** | | | | | |
| Tigecycline | 19 (63.33%) | 26 (63.41%) | 15 (62.5%) | 9 (40.9%) | 10 (41.67%) |
| **Cephalosporin** | | | | | |
| Cefepime | 11 (36.67%) | 28 (68.3%) | 17(70.83%) | 14 (63.64%) | 20 (80.33%) |
| **Carbapenem** | | | | | |
| Meropenem | 8 (26.67%) | 13 (31.7%) | 11 ((45.83%) | 6 (27.28%) | 13 (54.17%) |
| Imipenem | 10(33.34%) | 15 (36.59%) | 10 (41.67%) | 11 (50%) | 19 (79.17%) |
| Piperacillin/ Tazobactam | 6 (20%) | 19 (46.34%) | 11 (45.83%) | 12 (54.54%) | 16 (66.67%) |
| **Beta-lactamase** | | | | | |
| Ampicillin | 7 (23.34%) | 40 (97.57%) | 23 (95.83%) | 18 (81.81%) | 20 (83.33%) |
| **Fluoroquinolone** | | | | | |
| Norfloxacin | 13(43.33%) | 20(48.79%) | 9 (37.5%) | 12(54.54%) | 16 (66.67%) |
| **Glycopeptide** | | | | | |
| Vancomycin | | | | | 5(20.83%) |
| **Oxazolidinone** | | | | | |
| Linezolid | | | | | 8 (33.33%) |
| **Polymyxin E** | | | | | |
| Colistin | 24 (80%) | 35 (85.37%) | 9 (37.5%) | 17 (77.27%) | |

N indicates the total number of isolates.

and diminutive appearance. Regrettably, due to their lack of knowledge, these small blisters eventually developed into ulcers over time. In our study, out of seventy-five people, 41 (54.67%) were men and 34 (45.33%) were women (Table 1). It was found that men were more likely to develop DFU and eventually diabetic foot infections than women. This corresponds with previous studies conducted in Dhaka [26, 27], where the number of men suffering from DFU was twice that of women. This was true for other studies carried out in different parts of

**Table 7. Resistance status of bacterial isolates.**

| Isolated organisms | SDR N (%) | MDR N (%) |
|---|---|---|
| *Proteus spp.* | 9(30) | 17(56.67) |
| *Escherichia coli* | 4(18.18) | 18(81.81) |
| *Pseudomonas aeruginosa* | 4(16.7) | 14(58.33) |
| *Klebsiella pneumoniae* | 10(24.39) | 15(36) |
| *Staphylococcus aureus* | 9(37.5) | 15(62.5) |

Here, N indicates the total number of isolates that were resistant to either a single class of antibiotics (SDR) or multiple classes of antibiotics (MDR), and the percentage of such isolates is mentioned in parenthesis.

India [33–38], Iraq [39, 40], Ethiopia [4–7], Romania [20], China [13] and Kuwait [18]. This could be attributed to a variety of factors. There is a higher male involvement in outdoor activities compared to females [36], and this heightens susceptibility to foot trauma and harm [41–44]. Contrary to our findings, studies from India, Pakistan and Malaysia reported that women had a higher risk of having DFU [42, 44, 45]. A reason for this can be that in some areas of Africa and South Asia, particularly in some developing nations, prevailing socio-cultural beliefs restricted women from achieving higher levels of education compared to men. As a consequence, a disparity in knowledge emerged between the two genders [46].

The present study findings also showed that 70.67% of DFU patients were from urban areas, while 29.33% were from rural areas (Table 1). These findings corroborate to studies conducted in Pakistan [42] and Ethiopia [5, 14, 47]. The prevalence of diabetes might be higher among individuals living in urban areas due to their comparatively less active way of life, distinct dietary patterns, and an increased likelihood of being overweight or obese [25]. According to our study, a lack of physical activity significantly affects DFU (Table 2). About 29 (38.66%) patients maintained a physical exercise schedule, while 46 (61.33%) did not. The physical exercise schedule included walking, physical exercise, and domestic labor. A previous study conducted in Dhaka [27] showed that 70% of the DFU patients were from rural areas and 30% were from the urban areas, which was inconsistent with our findings. Their research findings indicated that a majority of individuals with DFU were consuming water from tube wells. A prior investigation carried out in Bangladesh has proposed a potential link between persistent exposure to arsenic from drinking water and the development of type 2 diabetes [48]. Another study conducted in India demonstrated that individuals from rural areas were disproportionately impacted. However, this could be attributed to the location of the medical college where the study took place, as it was situated in a rural setting [33]. Our study occurred in an urban setting, potentially influencing the higher representation of patients residing in urban areas.

In addition, the results of this study found that the age range of 46 to 50 years old has the highest prevalence of foot ulcers, which is 33.33% (Table 1). In previous studies conducted in Dhaka, the average age of the patients was similar [26, 27]. A study [33] conducted in India showed that DFU was not associated with the duration of diabetes and increasing age which was also observed in our study. Nonetheless, another study conducted in Dhaka [49] showed that the development of DFU was associated with patients aged 50 and over. Other studies [10, 13, 17, 18, 20, 34, 44, 45, 50, 51] showed that average age of DFU patients were 50–70 years. In contrast, a study conducted in Pakistan [42], revealed that diabetes patients aged 75 and older had a greater prevalence of foot ulcers.

The prevalence of DFU among employed patients was 50.67% in our study (Table 1). On the other hand, a case-control study showed people who were employed had a 65% lower risk of developing DFU than those who were unemployed [47]. In our study, a notable contradiction with previous research emerged as the majority of patients were employed. This discrepancy could potentially be explained by the finding that a significant proportion of our participants (61.33%) did not engage in regular physical exercise.

In this study, an overwhelming majority of the patients had Type II diabetes (97.33%) (Table 2). This corresponds with a study conducted in Bangladesh, which reported they found 45% of patients had Type II diabetes [49]. Another study conducted in Punjab found that people with type II diabetes are more likely to develop DFU than people with type I diabetes. Because Type II diabetes affects foot health more than type I diabetes [42]. This claim was supported by various other studies [5–7, 15, 20, 34, 35, 38, 50, 52]. However, a study conducted in 2022 [14] showed the number of type I diabetes patients (51.5%) suffering from DFU was slightly higher than the number of type II (48.4%) diabetes patients suffering from DFU.

In our study, the DFU patients had varying durations of diabetes, with the following distribution: 1–5 years (29.33%), 6–10 years (24%), 11–15 years (20%), 16–20 years (16%), 21–25 years (8%), and 26–30 years (2.66%) (Table 2). Notably, it was observed that the longer individuals had diabetes, the less likely they were to develop DFU. This observation aligns with a prior study conducted in Bangladesh, which reported a mean duration of 5.9–6.9 years for DFU patients [49], corroborating our findings. However, a conflicting study from Bangladesh [26] stated that all DFU patients in their sample had diabetes for durations ranging from 3 to 20 years, contrary to the mean duration observed in our study. In India, a study by Chavan (2018) indicated that 49.5% of patients with DFU had diabetes for more than 5 years, and 51.5% had diabetes for less than 5 years; however, they still emphasized that prolonged diabetes duration was a significant DFU risk factor [33]. Several studies have identified a threshold of approximately 10 years or more of diabetes as a significant factor in the development of DFU [4, 13, 14, 20, 34, 44, 45, 47, 52]. In 2014, Deribe [4] showed that diabetic patients with a duration over 10 years were 8.452 times more likely to develop DFU, and in 2018 Vibha [44] found that persons with diabetes for over 10 years were 3.7 times more likely to develop DFU compared to those with less than 5 years' duration contradicting our findings.

In our study, it was determined that 78.66% of people use insulin (Table 2). Numerous studies have consistently reported an elevated risk of DFU among individuals using insulin. For instance, research conducted in Northwest Ethiopia [5] discovered that 47% of insulin users developed DFUs. Similarly, another study by Chavan in 2018 [33] also emphasized the heightened risk associated with insulin use. Furthermore, a study conducted in 2020 [47] found that patients relying solely on insulin were 2.75 times more prone to DFUs when compared to those using oral hypoglycemic agents. Research from Iran [53] and Eastern Indonesia [51] corroborate these findings. Furthermore, insulin therapy may contribute to the severity of DFU through its potential involvement in triggering inflammatory responses within the body [54].

Our study also revealed that the prevalence of foot ulcers was greater among tobacco users (14.66%) than among other behavioral risk factors (tobacco products, alcohol, and others) (Table 2). A previous study [55] showed that DFUs are significantly associated with societal and behavioral factors such as disadvantaged socioeconomic status, inadequate healthcare access, limited education, solitary living, and tobacco use. Another study states tobacco users were 5.7 times more likely to develop DFU [33]. Furthermore, cigarette smoking has been validated as a risk element for amputations in cases of DFU. One potential explanation is that cigarette smoking elevates the production of reactive oxygen species, which in turn triggers oxidative stress within blood vessels and the nervous system. This process can give rise to inflammation, cellular harm, and apoptosis [56, 57].

In this study, patients who did proper foot care was around 53 (70.66%), and patients who did not take regular foot care was around 22 (29.33%) (Table 2). Patient foot care involved daily foot inspection, washing, and thorough drying. Patients who started the foot care process early on had less risk of major infection, and some study patients reported that they delayed the foot care due to ignorance but got better results at recovering after doing so. This correlation aligns with research conducted in different parts of Ethiopia, Kenya, and India [4, 6, 58, 59]. In a case-control study in Eastern Indonesia [51], diabetic individuals who regularly examined their feet experienced a 64% lower risk of diabetic foot ulcers (DFU), while those not engaging in foot self-care had a 2.52-fold higher likelihood of DFU development compared to active practitioners[5]. In contrast, a cross-sectional study carried out in Iraq has indicated that there is no significant correlation between the occurrence of DFU and daily foot care practices [40].

In our study, most patients with DFU (40%) didn't experience additional complications (Fig 2). Some reported high blood pressure (14%), kidney issues (9%), and gangrene (7%). A small proportion had eating disorders and low blood pressure (6%), while liver complications, vomiting, and high blood sugar affected around 3% of cases. Heart complications were observed in 2% of cases. Other issues like high cholesterol, lung problems, gastritis, thumb pain, swelling, diarrhea, and severe pain were rare, affecting only about 1% of patients. Numerous studies examining DFU reveal a diverse landscape of associated complications. While hypertension emerges as a recurrent factor across varied research, its prevalence spans a wide range from 42.46% to 58.4% [4, 6, 7, 13, 19]. Neuropathy exhibits substantial variability, ranging from 14.4% to 68.42%, highlighting the complexity of its occurrence [4, 6, 7, 13, 19, 34, 60]. Gangrene, a severe complication, varies notably between 1.6% and 48.33% across studies [13, 19, 22, 34, 61]. Other notable complications such as retinopathy and kidney issues present differing percentages, fluctuating between 7.9% and 51.61% [4, 6, 7, 13, 19, 22, 34, 60]. The prevalence of cardiac disease ranges from 5.3% to 34% based on various studies [13, 60]. Previous research indicated a 4% occurrence of gastritis [6] and a 28.89% prevalence of high cholesterol [13]. However, other less frequent complications, like liver issues, appear in only about 3% to 6% of cases [7, 13, 19]. These disparities may arise from factors like sample sizes, demographics, geographic locations, and diverse diagnostic criteria used in these studies. Overall, the risk factors for DFU are multifaceted and include aspects such as age, rural living, income, medical history, insulin use, diabetes complications, peripheral neuropathy, vascular disease, poor glycemic control, and certain lifestyle factors like smoking.

Within our study, DFU classified as Wagner grade 2 exhibited the highest proportion of patients with positive cultures, accounting for 33.33% of ulcers. Wagner grade 3 ulcers followed with 19.85%, as depicted in Fig 3. These findings were supported by previous studies [15, 34]. However, a previous study conducted in Bangladesh [26] showed that Wagner grade 3 ulcers were the most prevalent, followed by Wagner grade 2 ulcers. Additional research [6, 14, 20] conducted supports these findings. In parallel, other studies [10, 38] have identified that the highest prevalence of patients with infected DFU falls within the Wagner grades 3–5 category.

In the present study, *Klebsiella pneumoniae* (29%) was the most common isolated pathogenic bacteria. The other predominant isolated pathogenic bacteria were *Proteus spp.* (21.27%), *Pseudomonas aeruginosa* (17.02%), *Staphylococcus aureus* (17.02%) and *Escherichia coli* (15.60%). In addition, almost 20% of ulcers had no bacterial growth (Table 4). A previous study conducted in Dhaka [27] showed that *Bacillus cereus* (17%)was the most prevalent. This was followed by *Staphylococcus spp.* (13%) *Acinetobacter baumannii* (10%), *Enterococci spp.* (9%), *Klebsiella spp.* (7%) and *Citrobacter spp.* (2%). Another study conducted in Dhaka [26] revealed that gram-negative bacteria were the most prevalent (80%), followed by gram-positive bacteria (19.3%) and fungi (0.7%). The highest occurrence among gram-negative bacteria was *Pseudomonas spp.* (36 isolates), representing 48% of cases and a third of all isolates. Other identified organisms included *Proteus spp.* (33.3%), *Klebsiella spp.* (28%), *Escherichia coli* (14.7%), *Acinetobacter spp.* (6.6%), *Citrobacter spp.* (5.3%), *Serratia spp.* (1.3%), and *Providencia spp.* (1.3%). Among gram-positive bacteria, *Staphylococcus aureus* was the most common, constituting 21.3% of infections. From previous research, the prevalence of specific bacteria in DFU can be observed. Gram-negative bacteria, including *Klebsiella pneumoniae*, *Proteus spp.*, *Pseudomonas aeruginosa*, and *Escherichia coli*, are frequently associated with these ulcers. *Staphylococcus aureus*, a gram-positive bacterium, is also commonly found [10, 14, 15, 17, 18, 20, 21, 34–36, 38, 50, 52]. The exact proportions may vary across studies, but they collectively emphasize the importance of both gram-negative bacteria and gram-positive bacteria in DFU. *Staphylococcus aureus* exhibits the highest prevalence in most studies [14, 15, 17, 18, 20, 21, 34–36,

38, 50], while others emphasize *Pseudomonas aeruginosa* [10, 26], and *Escherichia coli* [52]. These findings align with the trends observed in the present study.

Out of all the samples analyzed in our study, 18.67% exhibited the presence of a single bacterial species, while 28% displayed the growth of two different bacterial species. Notably, 61.33% of the samples revealed the coexistence of multiple bacterial species, while 20% exhibited no bacterial growth at all (Table 5). In a study conducted in India [52], most of the patients had polymicrobial infection (55%), monomicrobial infection (37%) and (8%) had a sterile culture. Pus and tissue cultures predominantly revealed infections involving multiple microorganisms, while bone cultures predominantly exhibited infections caused by a single microorganism. Research carried out in Nigeria documented a 100% incidence rate of infection, encompassing a total of 97 isolated cases. Within the participant group, 28.0% experienced infections caused by a single microorganism, whereas 72.0% exhibited infections resulting from the presence of multiple microorganisms [17]. In Kuwait, a study revealed that polymicrobial infection was observed in 75% of the patients [50]. These observations align with the outcomes of our study.

In our research, it was found that all bacterial isolates, with the exception of *Proteus spp.*, were predominantly present in Wagner's grade 2 (Table 3). Interestingly, the prevalence of *Proteus spp.* remained consistent in both Wagner's grades 2 and 3. However, when it came to *Staphylococcus aureus* and *Pseudomonas aeruginosa*, the majority of isolates were only detected after reaching Wagner's grade 3. Conversely, for *Klebsiella pneumoniae*, most isolates were observed in Wagner's grade 1 following Wagner's grade 2. Notably, in Wagner's grade 4, *Escherichia coli* isolates emerged as the second most prevalent. A prior study conducted in 2003 [15] had previously noted that *Escherichia coli*, *Proteus spp.*, and *Staphylococcus aureus* bacterial isolates were most prevalent in Wagner's grade 2. Building upon these findings, a hypothesis suggests that during the initial stages of superficial wounds (Wagner 1 and 2), aerobic bacteria such as *Staphylococcus spp.*, *Streptococcus spp.*, and *Enterobacteriaceae* appear to be the primary culprits responsible for infections. However, as wounds progress to more severe stages (Wagner's grade 3 to 5) and evolve into ulcers, anaerobic bacteria tend to assume a more dominant role in causing infections, as previously observed in a study [62].

Our testing revealed significant antibiotic resistance patterns in *Escherichia coli* (Table 6). Notably, a substantial portion of these bacteria displayed resistance to azithromycin (86.36%), ampicillin (81.82%), and colistin (77.2%). Norfloxacin exhibited a 50% resistance rate, while moderate sensitivity was observed for streptomycin (77.2%), meropenem (72.7%), amikacin (68.1%), tigecycline (59%), and imipenem (50%). In contrast, a study in Dhaka [26] reported high amikacin resistance (63.6%) and moderate fluoroquinolone resistance (54.5%). Similarly, a study in India [34] found high resistance to azithromycin, amoxicillin, and colistin but high sensitivity to meropenem and amikacin. Tigecycline showed high resistance, unlike our study. Another Indian study [52] reported very high sensitivity to amikacin, meropenem, and imipenem. A Kuwaiti study [18] found minimal or no resistance to imipenem, meropenem, amikacin, piperacillin/tazobactam, and gentamicin in *Escherichia coli* strains. Norfloxacin resistance (50%) matched findings in another fluoroquinolone study [38]. Fluoroquinolone sensitivity varied in previous studies, with both high sensitivity [14, 17, 18, 20, 21, 35] and resistance [36, 50, 52] reported. In the case of *Klebsiella pneumoniae*, high resistance levels to ampicillin (97.5%) and colistin (85.3%) were observed (Table 6). Previous studies in Dhaka [26, 27] reported resistance to ampicillin, cephalosporins and fluoroquinolones, with cephalosporin resistance reaching around 70% and fluoroquinolone resistance at 66.7%. However, these studies showed different results for amikacin and aminoglycosides. Similar findings were reported in Iran [63], Kuwait [50] and India [10], where ampicillin exhibited high resistance to complete resistance in *Klebsiella pneumoniae*. Conversely, the Indian study [10] noted 100%

sensitivity to colistin. In our study, moderate resistance to cefepime (68.2%), tigecycline (60.5%), azithromycin (58.5%), and norfloxacin (51.2%) were observed. Notably, amikacin exhibited significant sensitivity (78%), while streptomycin (68.2%) and imipenem (63.4%) showed moderate sensitivity. In Ethiopia [14], all *Klebsiella pneumoniae* isolates were resistant to cefepime, with 60% resistance to imipenem and 50% resistance to amikacin. Conversely, a study in Kuwait [18] found minimal or negligible resistance to imipenem (0%), meropenem (0%), piperacillin/tazobactam (7%), and gentamicin (9%) in *Klebsiella spp*. In an Indian study [52], amikacin, imipenem, colistin, and cefepime were highly sensitive, while meropenem showed high resistance. For fluoroquinolones, some studies reported high resistance [10, 13], while most studies indicated high sensitivity to fluoroquinolones against *Klebsiella pneumoniae* isolates [14, 17, 18, 20, 35, 36, 50, 52].

In *Proteus spp*., substantial resistance to azithromycin (60%), colistin (73.3%), and tigecycline (63.3%) was detected (Table 6). Conversely, piperacillin-tazobactam (80%), ampicillin (76.6%), streptomycin (73.3%), meropenem (73.3%), imipenem (66.7%), cefepime (63.3%), amikacin (63.3%), and norfloxacin (56.6%) exhibited high sensitivity levels. In contrast to our study, a Dhaka study [26] reported high amikacin resistance (76%) and extensive resistance to fluoroquinolones (88%) in *Proteus spp*. For meropenem, complete susceptibility was observed in Italian and Kuwaiti studies [21, 50], whereas an Ethiopian study found 25% resistance [14], which aligns with our findings. Imipenem showed high sensitivity in previous studies [10, 14, 20, 35, 50] as did amikacin, with sensitivity ranging from 80% to 100% [14, 21, 50, 52]. Cefepime exhibited varying sensitivity levels (40%-100%) in different studies [10, 20, 52] although some reported high resistance [14]. Our study found high sensitivity to piperacillin-tazobactam, consistent with an Indian study [35], while other studies reported mixed results [14, 21]. Ampicillin sensitivity was also high in our study, similar to findings in Indian studies [10, 38] but contrasting with an Italian study [21] that reported high resistance (79.7%). Colistin resistance in *Proteus spp*. was high, aligning with other studies [10, 14]. Conversely, fluoroquinolones showed high sensitivity in previous studies [17, 20, 21, 36, 38, 50, 52].

In the case of *Pseudomonas aeruginosa* isolates, notable antibiotic resistance was observed, particularly against ampicillin (95.2%), cefepime (70.8%), azithromycin (66.6%), tigecycline (62.5%), and amikacin (50%) (Table 6). Conversely, sensitivity was seen to streptomycin (51.4%), colistin (62.5%), meropenem (54.1%), imipenem (58.3%), piperacillin-tazobactam (54.1%), and norfloxacin (62.5%). Ampicillin resistance was also high (96.7%) in a previous study [64]. In Dhaka, a study showed resistance to imipenem, amikacin, and fluoroquinolones [27]. Another Dhaka study reported cephalosporin resistance (66%-75%) [26]. High resistance to cefepime was observed in a study [14]. Contradictory findings were found in multiple studies regarding amikacin sensitivity, with many reporting high sensitivity [10, 14, 18, 20, 34–36, 38, 50, 52]. Colistin [65, 66], meropenem, imipenem, piperacillin-tazobactam [10, 18, 20, 34–36, 38, 50], and fluoroquinolones [10, 17, 18, 20, 21, 35, 36, 38, 50, 52] sensitivity was also high in numerous studies. However, one study in Ethiopia reported varying results, with piperacillin-tazobactam showing high resistance (66.7%), while meropenem (45.8%) and imipenem (37.5%) had lower resistance [14].

For *Staphylococcus aureus*, high sensitivity to vancomycin (79.17%) and norfloxacin (79.1%), along with moderate sensitivity to linezolid (66.67%) and tigecycline (63.8%) was detected (Table 6). This aligns with previous studies where vancomycin and linezolid were highly effective against gram-positive bacteria [10, 18, 20, 34, 50, 52, 63]. Again, *Staphylococcus aureus* strains in our study showed high resistance to cefepime (83.3%), ampicillin (83.3%), azithromycin (79.1%), and imipenem (79.1%). Moderate resistance to piperacillin-tazobactam (66.67%), amikacin (62.5%), streptomycin (58.3%), and meropenem (54.1%) was also observed. Similar high resistance in cephalosporins was reported in another study [66], and

ampicillin resistance was observed in previous studies [18, 20, 36]. However, other studies have shown high sensitivity to ampicillin in *Staphylococcus aureus* isolates [13, 50]. Azithromycin resistance was also found in previous studies [34, 35]. In contrast to our findings, piperacillin-tazobactam [34, 35], amikacin [34, 35, 38, 52] imipenem [64, 67, 68], and meropenem [64] showed high sensitivity in previous studies.

Most of the gram-negative and gram-positive bacteria were shown to be multidrug-resistant (Table 7). *Escherichia coli* exhibited the highest multidrug resistance at 81.8%, followed by *Pseudomonas aeruginosa* at 58.33%, *Staphylococcus aureus* at 62.5%, *Proteus spp*. at 56.67% and *Klebsiella pneumoniae* at 36%.In previous studies, high level of multidrug resistance was observed in *Pseudomonas aeruginosa* [14]. A previous study conducted in Dhaka [26] showed that gram-negative isolates possessed a high level of multidrug resistance. *Escherichia coli* (63.7%), *Pseudomonas spp*. (66.7%), *Proteus spp*.(76%) and *Klebsiella spp*.(62%) Another study also showed that all the isolates possessed a high level of multidrug resistance [14]. There, all the *Escherichia coli*, *Klebsiella pneumoniae* and *Proteus spp*. isolates were multidrug resistant. *Pseudomonas spp*. (91.6%) and *Staphylococcus aureus* (87.5%) isolates showed a high level of multidrug resistance.

In conclusion, DFU poses a significant threat to patients' physical, psychological, and financial well-being as DFU ranks among the costliest complications of diabetes [69]. Neglecting DFUs can result in the dire necessity of amputation, leading to distorted body image, loss of employment, dependency on others, increased healthcare costs, and psychological distress. The social impact is profound, with isolation, loss of social roles, and discrimination being common experiences [70, 71].

Physicians play a crucial role in educating patients about foot care and improving communication and education practices is essential [45]. Moreover, a shortage of foot care professionals and facilities exists in many parts of the world, emphasizing the need for effective diabetes management and secondary prevention of DFU [72].

Preventing the recurrence of DFU necessitates a multifaceted approach. This includes maintaining optimal blood sugar levels for effective diabetes control, scheduling regular professional foot care appointments every 1 to 3 months, and ensuring the use of properly fitting footwear that can alleviate plantar pressure and reduce the risk of DFU recurrence. Additionally, vigilant monitoring of skin temperature on the foot is essential to detect signs of inflammation promptly [73, 74]. Encouraging non-weight-bearing exercise as part of the management plan can contribute to overall patient well-being [75]. Furthermore, intensive glycemic control has been shown to significantly reduce the incidence of neuropathy and microvascular complications, including DFU. Smoking cessation is crucial, as smoking is associated with a higher DFU risk. Regular foot examinations by clinicians and referrals to specialists as necessary are vital components of care, and customized footwear and orthotics can provide relief and support for high-risk patients. A comprehensive and collaborative approach, coupled with patient education, is key to mitigating the recurrence of DFU [76–79]. As the saying goes, "prevention is better than treatment", and with the proper knowledge, awareness, and practices, patients can take proactive measures to safeguard their health and well-being from the ravages of DFU.

## Supporting information

**S1 Table. Wagner's classification of diabetic foot ulcers.**
(DOCX)

**S2 Table. Colony morphology of specific bacteria on selective media.**
(DOCX)

**S3 Table. Biochemical test interpretation for different bacteria.** * Hemolysis and coagulase test was performed on *Staphylococcus aureus* isolates for further confirmation of the isolates. <sup>&</sup> To confirm the identity of *Escherichia coli* isolates, they were subjected to additional subculturing on EMB Agar, and their colony morphology was observed. <sup>#</sup> To confirm the identity of *Proteus spp*. isolates, they were subjected to additional subculturing on Blood Agar, allowing for the observation of swarming colonies.
(DOCX)

**S4 Table. Primers and PCR conditions used in this study.**
(DOCX)

**S5 Table. List of antibiotics used in the experiment.**
(DOCX)

**S1 Fig. Agarose gel electrophoresis of PCR assay of *Pseudomonas aeruginosa* isolates.** Here, Lane M is 100 bp DNA marker, and Lane 1–3 are some positive samples at 956 bp. The other bands did not show the correct band size and therefore not included in our positive result.
(TIF)

**S2 Fig. Agarose gel electrophoresis of PCR assay of *Klebsiella pneumoniae* isolates.** Here, Lane M is 50 bp DNA marker, and Lane1-10 are some positive samples at 130 bp.
(TIF)

**S3 Fig. Agarose gel electrophoresis of PCR assay of *Escherichia coli* isolates.** Here, Lane M is 50 bp DNA marker, and Lane 1–10 are some positive samples at 585bp.
(TIF)

**S4 Fig. Agarose gel electrophoresis of PCR assay of *Staphylococcus aureus* isolates.** Here Lane (M) DNA is 50 bp marker, and Lane (1–10) are some positive samples at 108bp.
(TIF)

**S1 Raw images.**
(PDF)

## Acknowledgments

The team extends its gratitude to Jahid Hasan Tushar and Hasib Mahmud Rafi for their help in optimizing PCR conditions and other related tasks. We would also like to give a special thanks to Abdullah Al Taawab for his expertise in analyzing statistical information using SPSS 18.0 software.

## Author Contributions

**Conceptualization:** Poulomi Baral, Nafisa Afnan, Mohammed Muazzam Hossan.

**Data curation:** Mohammed Muazzam Hossan.

**Investigation:** Nafisa Afnan, Baby Akter, Shek Rabia Prapti.

**Methodology:** Poulomi Baral, Nafisa Afnan, Baby Akter.

**Software:** Nafisa Afnan.

**Supervision:** Fahim Kabir Monjurul Haque.

**Writing – original draft:** Poulomi Baral, Nafisa Afnan.

**Writing – review & editing:** Poulomi Baral, Nafisa Afnan, Maftuha Ahmad Zahra, Fahim Kabir Monjurul Haque.

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
