## [Decision Letter · Decision Letter 0]

21 Feb 2024

PONE-D-24-01408Bacteriological Analysis and Antibiotic Resistance in Patients with Diabetic Foot Ulcers in DhakaPLOS ONE

Dear Dr. Haque,

Thank you for submitting your manuscript to PLOS ONE. After careful consideration, we feel that it has merit but does not fully meet PLOS ONE’s publication criteria as it currently stands. Therefore, we invite you to submit a revised version of the manuscript that addresses the points raised by reviewer 1 during the review process.

We look forward to receiving your revised manuscript.

Kind regards,

Shailesh Kumar Shahi, PhD

Academic Editor

PLOS ONE

Reviewers' comments:

Reviewer's Responses to Questions

**Comments to the Author**

1. Is the manuscript technically sound, and do the data support the conclusions?

Reviewer #1: Partly

Reviewer #2: Yes

2. Has the statistical analysis been performed appropriately and rigorously? 

Reviewer #1: N/A

Reviewer #2: Yes

3. Have the authors made all data underlying the findings in their manuscript fully available?

Reviewer #1: Yes

Reviewer #2: Yes

4. Is the manuscript presented in an intelligible fashion and written in standard English?

Reviewer #1: Yes

Reviewer #2: Yes

5. Review Comments to the Author

Reviewer #1: Reviewer responses:

The authors carried out this study on an important and most studied disease like diabetes. They have conducted this research on people who have backgrounds in Bangladesh. The author's main study focus was on the diabetic ulcers and the bacterial isolates that are colonized, and the study was conducted more than the required relevance to socioeconomic and educational backgrounds. The authors give a lot of examples and more over frequently they are trying to justify their findings with the previous data and depend on survey studies conducted by the national and international diabetic organizations/ foundations. After reading, understanding and looking at their research results and discussions, I have a few major and minor comments on this manuscript.

Major concerns:

1. The authors could have taken up a larger demographics rather than a small number of cohorts. That could help to generate a wider bacterial colony to identify and test their antibiotic resistance in vitro conditions.

2. The authors could have focused on the host and the identified bacterial species and their specific relation in the background of diabetes, as the authors claimed they recruited people from the urban and rural areas which might include hygienic and unhygienic lifestyles might become non-specific hosts for several bacteria for a short period. Precisely identifying the diabetic ulcerative bacterial populations will provide a valuable clue for the treatment.

3. The authors seem to have the same kind of growth media to culture different species of bacteria such as gram-positive and gram-negative and also we cannot rule out the bacterial adaptations to the available nutrients that potentially cause response or resistance to different antibiotics.

4. I would suggest the authors conduct more deliberate experiments to see the bacterial responses to an antibiotic, the current study involved basic experimental settings instead of deep investigations.

Minor concerns:

1. The authors have missed a great option to show their all data and importantly they have limitedly presented their results and focused to discuss on the already available data from various sources to justify their study.

2. The figures have a very poor resolution and are not able to read the Figure 1 contents at all.

Impression:

This study will help as a little reference for further studies whoever wants to investigate deep into the diabetic ulcerative bacteria in connection with antibiotic resistance.

Reviewer #2: Authors did an excellent job. The manuscript is well written. It has good research with innovative aim.

I will recommend for publication. Some minor grammatical mistakes are in the manuscript that needs to correct before the publication.

6. PLOS authors have the option to publish the peer review history of their article (what does this mean?). If published, this will include your full peer review and any attached files.

Reviewer #1: No

Reviewer #2: **Yes**

---

## [Author Response · Author response to Decision Letter 0]

29 Feb 2024

Reviewer #1:

Reviewer responses:

The authors carried out this study on an important and most studied disease like diabetes. They have conducted this research on people who have backgrounds in Bangladesh. The author's main study focus was on the diabetic ulcers and the bacterial isolates that are colonized, and the study was conducted more than the required relevance to socioeconomic and educational backgrounds. The authors give a lot of examples and more over frequently they are trying to justify their findings with the previous data and depend on survey studies conducted by the national and international diabetic organizations/ foundations. After reading, understanding and looking at their research results and discussions, I have a few major and minor comments on this manuscript.

Major concerns:

1. The authors could have taken up a larger demographics rather than a small number of cohorts. That could help to generate a wider bacterial colony to identify and test their antibiotic resistance in vitro conditions. 

Author’s Response: 

We value the feedback from the reviewers. Despite the constraints posed by limited resources, facilities, and the challenges associated with conducting clinical studies in Bangladesh, we made concerted efforts to optimize our sample size. It is important to note that our study is only the third of its kind in Bangladesh (1,2), with the most recent one conducted nearly a decade ago (1). This scarcity of research underscores the significance and timeliness of our contribution to the existing body of knowledge in this domain. While we do recognize the potential benefits of a larger population and the inclusion of a broader range of bacterial species to enhance thoroughness, nonetheless, our study serves as a catalyst for further, extensive research on this topic.

2. The authors could have focused on the host and the identified bacterial species and their specific relation in the background of diabetes, as the authors claimed they recruited people from the urban and rural areas which might include hygienic and unhygienic lifestyles might become non-specific hosts for several bacteria for a short period. Precisely identifying the diabetic ulcerative bacterial populations will provide a valuable clue for the treatment.

Author’s Response: 

We appreciate the reviewer’s comments and acknowledge them. An unhygienic lifestyle can heighten the susceptibility of sores to infections. Nevertheless, due to the prolonged persistence of infections, the patients were hospitalized. All participants in our study were hospitalized, ensuring adherence to rigorous hygienic guidelines. Moreover, our study centered on patients from Ibrahim Diabetic Foot Care Hospital and Shaheed Suhrawardy Medical College and Hospital, institutions renowned for their advanced diabetic treatment facilities. Samples were obtained from deep pus, and the infections were categorized based on Wagner's grade, indicating active infections. This strongly implies that the identified bacteria are the causative agents of these infections. Our findings guided physicians in prescribing antibiotics.

Nevertheless, we recognize a limitation in our focus on hosts. Despite our efforts to prioritize patients, time constraints and follow-up challenges hindered a more in-depth exploration of host-bacteria relationships. We concede that a more extensive investigation into the host factors could have yielded additional insights. Due to this, our study emphasized analyzing bacterial species and their antibiotic resistance profiles.

3. The authors seem to have the same kind of growth media to culture different species of bacteria such as gram-positive and gram-negative and also we cannot rule out the bacterial adaptations to the available nutrients that potentially cause response or resistance to different antibiotics.

Author’s Response: 

We acknowledge the reviewer's feedback. In our study, only one gram-positive bacteria, specifically Staphylococcus aureus, was isolated. The standard mannitol salt agar (MSA) was employed as the selective medium for its cultivation. For other gram-negative bacterial species, we utilized well-established culture media: Cetrimide Agar for Pseudomonas aeruginosa, HiCrome KPC Agar for Klebsiella pneumoniae, HiCrome UTI Agar for Escherichia coli (confirmed later using EMB Agar), and MacConkey Agar for Proteus spp. These culture media are routinely employed in the growth of their respective bacteria. Information about the culture media used for growing these bacteria is provided in S2 Table. 

It is noteworthy that none of our culture media incorporated antibiotic supplements, mitigating the likelihood of bacterial adaptations to available nutrients potentially influencing responses or resistance to different antibiotics.

4. I would suggest the authors conduct more deliberate experiments to see the bacterial responses to an antibiotic, the current study involved basic experimental settings instead of deep investigations.

Author’s Response: 

We appreciate the reviewer's suggestion. The in vitro antibiotic testing of the isolates was conducted in triplicate. The primary focus of our research was to assess resistance of bacterial isolates to the recommended antibiotic dosages. Our study did not delve into the effects of antibiotics during follow-up; this was beyond the scope of our investigation.

Minor concerns:

1. The authors have missed a great option to show their all data and importantly they have limitedly presented their results and focused to discuss on the already available data from various sources to justify their study. 

Author’s Response:

We appreciate the reviewer's comments. Considering the reviewers concern we have made adjustments to ensure a greater focus on our results. All the data in the revised manuscript has been thoroughly presented. This modification aims to prevent our findings from being overshadowed by data from other studies.

2. The figures have a very poor resolution and are not able to read the Figure 1 contents at all.

Author’s Response:

We appreciate the feedback from the reviewers. The photos have been formatted in accordance with PLOS One guidelines. However, we acknowledge that the resolution may have been compromised during the conversion of the photos into a PDF file. We concur with the observation that the contents of Figure 1 are challenging to read, and consequently, we have implemented modifications to enhance its legibility.

Reviewer #2: 

Authors did an excellent job. The manuscript is well written. It has good research with innovative aim.

I will recommend for publication. Some minor grammatical mistakes are in the manuscript that needs to correct before the publication.

Author’s Response:

We are appreciative of the reviewer’s comments. We have thoroughly rechecked our manuscript and corrected all the grammatical mistakes. 

References:

1. Karmaker M, Sanyal SK, Sultana M, Hossain MA. Association of bacteria in diabetic and non-diabetic foot infection - An investigation in patients from Bangladesh. J Infect Public Health [Internet]. 2016;9(3):267–77. Available from: http://dx.doi.org/10.1016/j.jiph.2015.10.011

2. Paul S, Barai L, Jahan A, Haq JA. A Bacteriological Study of Diabetic Foot Infection in an Urban Tertiary Care Hospital in Dhaka City. Ibrahim Med Coll J. 2009;3(2):50–4.

---

## [Decision Letter · Decision Letter 1]

21 Mar 2024

Bacteriological Analysis and Antibiotic Resistance in Patients with Diabetic Foot Ulcers in Dhaka

PONE-D-24-01408R1

Dear Dr. Haque,

We’re pleased to inform you that your manuscript has been judged scientifically suitable for publication and will be formally accepted for publication once it meets all outstanding technical requirements.

Kind regards,

Shailesh Kumar Shahi, PhD

Academic Editor

PLOS ONE

Additional Editor Comments (optional):

Dear Dr. Fahim Kabir Monjurul Haque,

Thank you very much for properly revising the manuscript. I would appreciate it if the resolution of the figures could be improved. High-quality visuals are crucial to effectively convey our findings and ensure clarity for the readers.

Additionally, I recommend a thorough review and correction of the English grammar and language usage throughout the manuscript. I believe that these adjustments will significantly enhance the overall quality of the publication.

Best Wishes,

Shailesh Shahi

Reviewers' comments:

Reviewer's Responses to Questions

**Comments to the Author**

1. If the authors have adequately addressed your comments raised in a previous round of review and you feel that this manuscript is now acceptable for publication, you may indicate that here to bypass the “Comments to the Author” section, enter your conflict of interest statement in the “Confidential to Editor” section, and submit your "Accept" recommendation.

Reviewer #1: All comments have been addressed

Reviewer #2: All comments have been addressed

2. Is the manuscript technically sound, and do the data support the conclusions?

Reviewer #1: Partly

Reviewer #2: Yes

3. Has the statistical analysis been performed appropriately and rigorously? 

Reviewer #1: Yes

Reviewer #2: Yes

4. Have the authors made all data underlying the findings in their manuscript fully available?

Reviewer #1: Yes

Reviewer #2: Yes

5. Is the manuscript presented in an intelligible fashion and written in standard English?

Reviewer #1: Yes

Reviewer #2: Yes

6. Review Comments to the Author

Reviewer #1: (No Response)

Reviewer #2: (No Response)

7. PLOS authors have the option to publish the peer review history of their article (what does this mean?). If published, this will include your full peer review and any attached files.

Reviewer #1: **Yes: **Chakrapani Vemulawada

Reviewer #2: **Yes: **Avinash Singh

---

## [Editor Report · Acceptance letter]

30 Apr 2024

PONE-D-24-01408R1 

PLOS ONE

Dear Dr. Haque, 

I'm pleased to inform you that your manuscript has been deemed suitable for publication in PLOS ONE. Congratulations! Your manuscript is now being handed over to our production team.

Kind regards, 

on behalf of

Dr. Shailesh Kumar Shahi 

Academic Editor

PLOS ONE